# Effect of a brief motivational interview and text message intervention targeting tobacco smoking, alcohol use and medication adherence to improve tuberculosis treatment outcomes in adult patients with tuberculosis: a multicentre, randomised controlled trial of the ProLife programme in South Africa

Goedele Louwagie [1,2] Mona Kanaan,[3] Neo Keitumetse Morojele,[4,5] Andre Van Zyl [1] Andrew Stephen Moriarty [6] Jinshuo Li [3] Kamran Siddiqi [6] Astrid Turner,[2] Noreen Dadirai Mdege,[3] Olufemi Babatunde Omole,[7] John Tumbo,[8] Max Bachmann [9] Steve Parrott,[3] Olalekan A Ayo-Yusuf[2,10]

For numbered affiliations see end of article.

**Correspondence to**
Dr Goedele Louwagie;
Goedele.louwagie@up.ac.za

## ABSTRACT

**Objective** To investigate the effectiveness of a complex behavioural intervention, ProLife, on tuberculosis (TB) treatment success, medication adherence, alcohol use and tobacco smoking.

**Design** Multicentre, individual, randomised controlled trial where participants were assigned (1:1) to the ProLife intervention or usual care.

**Setting** 27 primary care clinics in South Africa.

**Participants** 574 adults starting treatment for drug-sensitive pulmonary TB who smoked tobacco or reported harmful/hazardous alcohol use.

**Interventions** The intervention, delivered by lay health workers (LHWs), consisted of three brief motivational interviewing (MI) sessions, augmented with short message service (SMS) messages, targeting medication adherence, alcohol use and tobacco smoking.

**Outcome measures** The primary outcome was successful versus unsuccessful TB treatment at 6–9 months, from TB records. Secondary outcomes were biochemically confirmed sustained smoking cessation, reduction in the Alcohol Use Disorder Identification Test (AUDIT) score, improved TB and antiretroviral therapy (ART) adherence and ART initiation, each measured at 3 and 6 months by questionnaires; and cure rates in patients who had bacteriology-confirmed TB at baseline, from TB records.

**Results** Between 15 November 2018 and 31 August 2019, 574 participants were randomised to receive either the intervention (n=283) or usual care (n=291).

## Strengths and limitations of this study

► The use of motivational interviewing combined with short text messaging to address the effect of multiple risk behaviours (smoking, drinking and poor adherence) on tuberculosis treatment outcomes is a novel and much needed intervention.

► Our study design was strong: this was a multisite, individually randomised controlled trial with a large sample size and a high follow-up rate for the primary outcome.

► We used validated measurement tools; furthermore, data analysis and primary outcome assessment were blinded, thereby limiting measurement bias.

► However, the study was underpowered for secondary outcomes, and low intervention uptake may have diluted any potential intervention effects.

TB treatment success rates did not differ significantly between intervention (67.8%) and control (70.1%; OR 0.9, 95% CI 0.64% to 1.27%). There was no evidence of an effect at 3 and 6 months, respectively, on continuous smoking abstinence (OR 0.65, 95% CI 0.37 to 1.14; OR 0.76, 95% CI 0.35 to 1.63), TB medication adherence (OR 1.22, 95% CI 0.52 to 2.87; OR 0.89, 95% CI 0.26 to 3.07), taking ART (OR 0.79, 95% CI 0.38 to 1.65; OR 2.05, 95% CI 0.80 to 5.27) or AUDIT scores (mean score difference 0.55, 95% CI −1.01 to 2.11; −0.04, 95% CI −2.0 to 1.91)

and adjusting for baseline values. Cure rates were not significantly higher (OR 1.16, 95% CI 0.83 to 1.63).
**Conclusions** Simultaneous targeting of multiple health risk behaviours with MI and SMS using LHWs may not be an effective approach to improve TB outcomes.
**Trial registration number** ISRCTN62728852.

## INTRODUCTION

Tuberculosis (TB) is among the most common chronic infectious diseases in the world today. In 2019, 1.4 million deaths worldwide were attributed to TB, and the majority of these occurred in low-income and middle-income countries (LMICs).[1] Not only has South Africa one of the highest TB burdens in the world but also it is faced with high TB treatment interruption and loss to follow-up rates. It also has a high prevalence of HIV coinfection in patients with TB and a relatively high mortality in these coinfected patients.[1] Studies of interventions to advance the goal of ending the TB epidemic and improving treatment outcomes are therefore research priorities in South Africa and in other LMICs.[2]

Mortality and morbidity from TB are strongly associated with health risk behaviours, particularly smoking and hazardous or harmful alcohol use, both of which are prevalent and often co-occur in patients with TB.[3–10] Strategies are also required to improve TB medication adherence in patients with TB and adherence to TB medication and antiretroviral therapy (ART) in patients coinfected with TB and HIV, both of which may be negatively influenced by excessive alcohol use.[1] There is very limited research on how to concurrently tackle these three risk behaviours—namely, smoking, harmful alcohol use and poor medication adherence—in patients with TB, particularly in LMICs.

Motivational interviewing (MI) has been shown to support reduced drinking, smoking cessation in patients with TB, and TB treatment and/or ART medication adherence.[11–13] MI interventions can be effectively delivered by lay health workers (LHWs).[14] The more widespread use of LHWs and the increased use of mobile health (mHealth) digital technologies represent promising ways to increase the scalability of MI interventions. Indeed, the WHO has called for researchers to capitalise on advances in mobile phone technology, network coverage and the increased use of common and widely available digital technologies (including the mobile phone short message service (SMS)) to improve TB care.[15] There is evidence that mHealth technologies can have modest beneficial effects on a range of health outcomes, including medication adherence.[16 17] Mobile phone messaging also shows a modest effect in improving TB treatment success rates.[18 19] The evidence is, however, stronger for two-way messaging and interactive systems for which smart phones are required.[18] These are often not available to patients with TB in Africa.[20]

A limitation of existing MI and mHealth interventions is that they have been studied in the context of modifying a single lifestyle factor. Integrated interventions are likely to be better accepted and more effective than multiple interventions targeting different health risk factors.[21 22] In the case of TB, there is a need for an intervention that has the flexibility to target multiple lifestyle factors as appropriate and in line with patient preferences. This could be achieved through increased integration of TB and non-communicable disease services.[23]

Recent re-engineering of primary healthcare in South Africa has seen the introduction of municipal ward-based primary healthcare outreach teams of community health workers (CHWs). CHWs work in an integrated, team-based manner, supported by nurses, and take responsibility for health education and promotion, counselling and support for a range of health conditions.[24 25] Task shifting in this context has been shown to improve population health in LMICs,[26] and these teams can be trained and supported to take responsibility for TB/HIV care.[27] Integrated interventions could be implemented within this framework in a feasible and scalable way to improve outcomes for patients with TB across South Africa and beyond.

Building on previous successes with MI and mHealth interventions, we developed a complex behavioural intervention (ProLife) comprising MI-based counselling and SMS, targeting three lifestyle risk behaviours for poor TB outcomes (smoking, hazardous/harmful alcohol consumption and poor medication adherence) and delivered by LHWs. We then conducted a randomised controlled trial (RCT) to assess the effectiveness of the ProLife intervention on improving TB treatment outcomes, smoking abstinence, reducing alcohol consumption, and improving adherence to TB and ART medication compared with usual care. The cost-effectiveness of the intervention was also assessed, but only the costing results will be presented in this paper.

## METHODS

### Study design and participants

This was a prospective, two-arm, multicentre, individual RCT which took place across 27 primary care clinics in three districts in South Africa (Lejweleputswa in the Free State province, Bojanala in the North West province and Sedibeng in Gauteng province). Adult patients (18 years or older) were eligible for the study if they had drug-sensitive pulmonary tuberculosis (PTB) and were initiating TB treatment or had been on TB treatment for less than a month for this treatment episode (both 'new' and 'retreatment patients'). They had to be tobacco smokers (defined as smoking daily or non-daily in the last 4 weeks on the Global Adult Tobacco Survey questionnaire)[28] and/or hazardous/harmful drinkers who were not alcohol dependent (Alcohol Use Disorder Identification Test (AUDIT) score of ≥8 for men or ≥7 for women but <20).[29] They also had to have access to a mobile phone and understand one of the four languages used for the trial (English, IsiZulu, SeSotho and Setswana). Potential participants were recruited consecutively at the

participating clinics between 15 November 2018 and 31 August 2019. Trained field workers identified those interested in the study and screened them for eligibility. If eligible and willing to be enrolled into the trial, written informed consent was obtained.[30]

## Randomisation and blinding

Patients were centrally randomised (1:1) to the ProLife intervention or control group using a randomised sequence generator by the trial statistician (MK) who was blinded to the arm allocation. We used block randomisation with varying block sizes stratified by clinic so as to achieve equal numbers in intervention and control groups within each clinic. Fieldworkers used sequentially numbered, sealed, opaque envelopes to allocate participants to intervention or control. ProLife involved a complex behavioural intervention; therefore, LHWs and participants could not be blinded to the intervention. However, the determination of the primary outcome was done by the TB nurses who were blinded to the intervention status of the participants, based on routinely collected data. The statistician (MK) was blinded to the intervention or control arm allocation of participants during the analysis.

## Intervention and procedures

The ProLife intervention was developed based on a conceptual framework, following a review of pre-existing evidence.[31] This framework assumed that smoking cessation, reducing harmful alcohol use and improved adherence to TB and HIV treatment would result in improved TB treatment outcomes.[30] The intervention consisted of three brief MI counselling sessions, lasting 15–20 min, 1 month apart, delivered by trained LHWs at their TB clinic. The first MI session took place immediately or shortly after the randomisation and involved prioritisation and agenda setting, wherein the participant determined which factor should be prioritised (either a plan to quit tobacco smoking or to reduce or quit drinking, or to deal with barriers relating to ART or TB medication adherence). The second and third sessions built on the previous one until all relevant behavioural problems had been addressed. These sessions were reinforced with follow-up SMS text messages, two times per week over 12 weeks.[30] Study patients received 10 TB-related messages followed by seven alcohol reduction-related and/or seven smoking cessation-related messages, as appropriate. Messages were aimed at giving information and augmenting motivation or behavioural skills (we refer to the feasibility paper for more details).[31] Applicable SMS messages were automatically activated after the first MI had taken place. Thereafter, remaining messages were delivered even if the participant did not attend the second or third MI session.

Participants randomised to the ProLife intervention also received the same 'usual care' as those in the control group. The control group received the usual care and routine treatment and support offered to patients with TB in South Africa, which vary by district but include health education, dietetic input, social support, point of care biochemical testing, and HIV testing with pretest and post-test HIV test counselling.

Data were collected at baseline and 3 and 6 months and were recorded by fieldworkers equipped with mobile phones with the ProLife mobile data collection application (built with CommCare)[32] installed. They used a standardised electronic case report form (CRF) and followed standard operating procedures to ensure quality. Details of data collection, protection and storage procedures were reported elsewhere.[30]

## Patient and public involvement

Patients or the public were not involved in the design, conduct, reporting or dissemination plans of our research.

## Outcomes

### Primary outcome

The primary outcome of TB treatment success at 6–9 months of follow-up (depending on when it was recorded) was as per the WHO definitions adopted in South Africa,[10] that is, either successful treatment (cured or treatment completed) or failed treatment, death, acquired drug resistance, loss to follow-up (defined as treatment interruption of more than 2 months) or outcome not evaluated. It was measured using the routinely collected TB treatment outcomes in patients' individual files.

### Secondary outcomes

For those participants with bacteriologically confirmed PTB at baseline (either sputum acid-fast bacilli-positive, culture positive or GeneXpert-positive PTB), sputum conversion at the end of treatment ('cure rate') was measured as a secondary outcome.[10] Continuous smoking abstinence was assessed at 3 and 6 months of follow-up in those participants who were current cigarette smokers at baseline. It was defined as having quit smoking completely and a self-report of not smoking more than five cigarettes from the start of the study, in addition to a negative biochemical test (exhaled carbon monoxide (CO) <7 ppm).[33 34] Changes in alcohol consumption were computed using the AUDIT questionnaire scores measured at 3 and 6 months of follow-up in those participants who were hazardous/harmful drinkers at baseline.

HIV-positive participants were asked about ART status at baseline and 3 and 6 months using standardised questions on the CRF and change in ART status as measured at the two follow-up times.

TB and ART medication adherence was measured using modified versions of the AIDS Clinical Trials Group Adherence Questionnaire, a validated tool for measuring adherence specifically to ART.[35] Adherence was measured using an adherence index calculated by the formula (using the 4-day recall): [total number of doses taken/total number of doses prescribed]×100. Patients with at least 95% adherence were classed as having optimal adherence, and those with less than 95% were classed as



having low (or suboptimal) adherence. This was assessed at 3 and 6 months.

During COVID-19 lockdown (in the second term of 2020), we switched to telephonic follow-up of participants using a shortened questionnaire whereby only strictly needed information for the measurement of outcomes was inquired about.

### Training and intervention fidelity monitoring

The training and intervention fidelity monitoring is described in more detail in previous papers.[30 31] In brief, 18 LHWs, 3 district coordinators and 1 research assistant who focused on counselling supervision underwent MI training over 5 days. LHWs completed a postsession semi-structured form onto which they indicated the extent to which they implemented each element of MI, as well as their general qualitative impressions of that particular session. In addition, we assessed MI intervention fidelity based on ratings of the counsellors' recorded MI sessions, as described further and in footnotes to the online supplemental table 4. SMS-message delivery was also assessed.

### Economic evaluation

The ProLife intervention costs consisted of the costs of training and the delivery of the ProLife intervention, including relevant personnel involvement (trainers and LHWs), materials used, travel, accommodation and refreshments, and digital infrastructure for the intervention. These were estimated based on research team records. Usual care costs consisted of TB medication costs, biochemical investigations and ART costs if applicable. These were estimated based on information obtained through routine records. The country-specific version of EuroQol with five dimensions and three levels of response categories (EQ-5D-3L) for South Africa was administered to participants at baseline and 3 and 6 months of follow-up to measure health-related quality of life.[36 37]

### Statistical analysis

The sample size was estimated at 696 in total (348 participants per arm) to detect a 10% difference in TB treatment success rates (0.86 vs 0.76) in the ProLife arm (intervention) versus the control arm with 80% power, a significance level of 0.05% and 25% attrition rate. The assumed success rates in the control group were based on actual success rates in patients with TB in the studied provinces obtained from TB managers at the time of the grant application for this study.

We summarised baseline data descriptively by trial arm. For the primary outcome, we conducted statistical analysis on an intention-to-treat basis. We used binary logistic regression to compare the main outcome (TB treatment success rate) between the intervention and the usual care arm. Where treatment outcome data were missing, the outcome was coded as unsuccessful. TB treatment outcomes recorded by the TB nurse were taken on face value as inconsistencies in the dates of bacteriological

results did not permit us to verify the correctness of the nurse assessment. We carried out similar statistical analyses for the secondary outcomes with appropriate regression techniques. For the reduction in harmful or hazardous drinking, we used linear regression to estimate the difference in total AUDIT score between control and intervention groups accounting for the baseline AUDIT score as covariate. Separate analyses at 3 and 6 months were performed.

For our main analyses, we adjusted for baseline characteristics if these differed between trial arms at baseline. The covariates that we controlled for in each model are specified when a model is presented. The statistical packages Stata[38] and R[39] were used to carry out the analyses, with a p value of <0.05 considered statistically significant.

The validated Motivational Interviewing Treatment Integrity (MITI) coding tool V.4.2.1 was used to assess MI intervention fidelity.[40] The coding entailed making 'global ratings' (on four dimensions: cultivating change talk, softening sustain talk, partnership and empathy) and 'behaviour' counts (with respect to the items giving information, persuade, persuade with permission, question, simple reflection, complex reflection, affirm, seeking collaboration, emphasising autonomy and confront). A score was assigned to each of these items, and the scores were compared against the competency and proficiency thresholds that are specified in the MITI manual.

For the analysis of the costs, all costs were collected in South Africa Rand (ZAR) except for the data management system subscription. Results are presented in both ZAR and US dollar using the 2019 Organisation for Economic Co-operation and Development (OECD) exchange rate (US$1=14.448 rand).[36] No South African specific valuation set was available for EQ-5D-3L. The valuation set of Argentina, based on a Visual Analogue Scale, was used to derive utility values, because the Gross Domestic Product (GDP) per capita in international dollars was the closest between the two countries at the time of analysis.[37 41] Quality-adjusted life years (QALYs) were derived from the utility values at the time points by calculating the area under the curve.[42] No missing data imputation was performed.

Data were stored in the institutional data repository at Sefako Makgatho Health Sciences University. Data will be embargoed until 30 June 2023 after which they will be freely accessible.[43]

## RESULTS

### Participant enrolment and follow-up

A total of 2099 patients with TB were screened for eligibility, out of which 574 consenting and eligible participants were randomised: 291 to control and 283 to intervention. Trial recruitment was terminated on 31 August 2019 before the planned sample size was reached because of budget and time constraints. In the intervention arm, 227 (80.2%) participants completed the first MI (MI 1) session; 199 (70.3%) completed MI 2; and

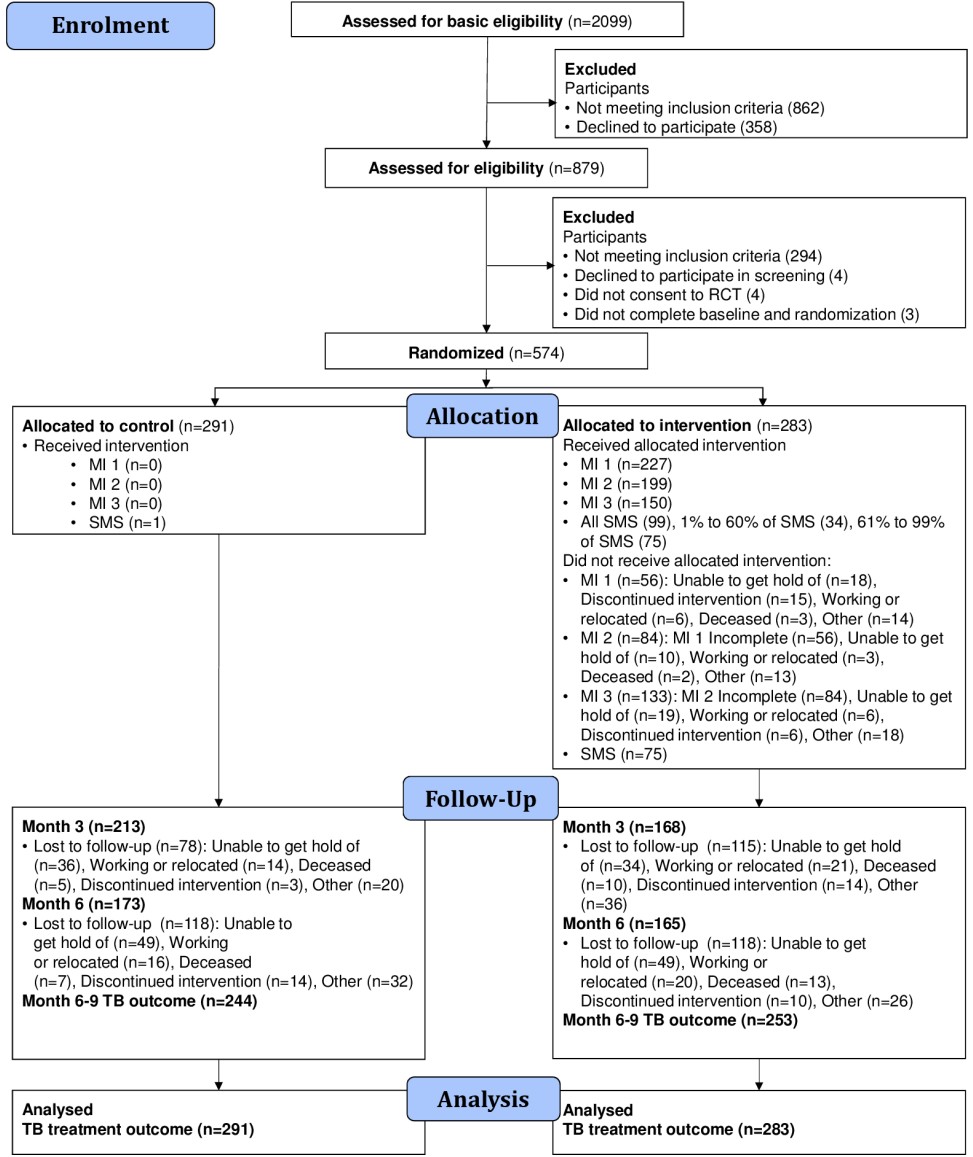

**Figure 1** Consolidated Standards of Reporting Trials flow diagram. RCT, randomised controlled trial; SMS, short message service; TB, tuberculosis.

150 (53.0%) completed MI 3. In the intervention arm, at least one message was delivered to 208 (73.5%) participants, while 99 (35.0%) received all messages. Of those randomised to the control and intervention groups, the primary outcome was recorded in 244 (83.8%) and 253 (89.4%) participants, respectively (figure 1).

## Baseline participant characteristics of the intervention and control arms

Baseline characteristics were distributed similarly in the intervention and control arms for most variables but with some imbalances in educational level. A total of 513 (91.3%) participants were new patients with TB, 129 (22.5%) women, and nearly all had PTB (International Classification of Diseases-10 A15) without extra-PTB manifestations (553, 98.9%). About half of the participants were HIV positive (305, 53.2%), of whom 204 (65.4%) were on cotrimoxazole and 257 (82.4%) were

on ART (table 1). Details of marital status, employment, wealth, depression status and comorbidities are presented in online supplemental table 1.

There were 372 current smokers (298 daily, 74 less than daily). Seventy-eight participants (26.8 %) in the control arm were dual smokers *and* drinkers compared with 114 (40.3 %) in the intervention arm. In the control arm, 110 (37.8 %) were hazardous/harmful drinkers only and 103 (35.4 %) were smokers only, compared with 92 (32.5%) and 77 (27.2), respectively, in the intervention arm (table 2). More details of smoking and drinking history, forms of tobacco use, addiction and quit attempts are presented in online supplemental table 2.

## Primary outcome

Overall, 396 (70%) of participants were classified as treated successfully (treatment completed or cured). The remainder either interrupted treatment, failed treatment,

**Table 1** Baseline descriptive socioeconomic statistics and clinical characteristics by study arm

| | Control (N=291) n (%)* | Intervention (N=283) n (%)* | Total (N=574) n (%)* |
|---|---|---|---|
| Age (years), mean (SD) | 39.37 (12.60) | 38.56 (11.15) | |
| Female sex | 69 (23.7) | 60 (21.2) | 129 (22.5) |
| **Education** | | | |
| No education | 7 (2.4) | 5 (1.8) | 12 (2.1) |
| Grades 1–5 | 23 (7.9) | 20 (7.1) | 43 (7.5) |
| Grades 6–7 | 32 (11.0) | 35 (12.4) | 67 (11.7) |
| Grades 8–11 | 96 (33.0) | 128 (45.2) | 224 (39.0) |
| Grade 12 | 87 (29.9) | 70 (24.7) | 157 (27.4) |
| Higher | 24 (8.2) | 8 (2.8) | 32 (5.6) |
| Declined to answer† | 22 (7.6) | 17 (6.0) | 39 (6.8) |
| **TB patient category** | | | |
| New patient | 264 (92.3) | 249 (90.2) | 513 (91.3) |
| Relapse | 10 (3.5) | 9 (3.3) | 19 (3.4) |
| Retreatment after default | 9 (3.1) | 14 (5.1) | 23 (4.1) |
| Retreatment after failure | 1 (0.3) | 2 (0.7) | 3 (0.5) |
| Other | 2 (0.7) | 2 (0.7) | 4 (0.7) |
| TB site of disease pulmonary only (International Classification of Diseases, ICD-10 A15) | 281 (98.9) | 272 (98.9) | 553 (98.9) |
| TB sputum smear, Gene XPert or culture result available (N) | 236 | 227 | 463 |
| At least one sputum smear, Gene XPert or culture result positive | 208 (88.1) | 195 (85.9) | 403 (87.0) |
| **HIV status** | | | |
| Negative | 118 (40.7) | 125 (44.2) | 243 (42.4) |
| Positive | 163 (56.2) | 142 (50.2) | 305 (53.2) |
| Unknown | 9 (3.1) | 16 (5.7) | 25 (4.4) |
| **HIV-positive patients** | | | |
| Using cotrimoxazole | 104 (63.8) | 100 (67.1) | 204 (65.4) |
| Using antiretroviral therapy | 139 (85.3) | 118 (79.2) | 257 (82.4) |

*Frequencies and (percentages) are presented unless otherwise stated.
†More variables with the option 'declined to answer' are listed in online supplemental table 1.

developed drug resistance, were transferred out or had an unknown treatment outcome (online supplemental table 3). The percentage of successful TB treatment did not differ significantly between the control and intervention arm (70.1% vs 67.8%), OR for successful TB treatment of 0.90 (95% CI 0.64 to 1.27) comparing the intervention arm to the control arm, and was similar to adjusted ORs (tables 3 and 4).

### Secondary outcomes
#### Cure rates
Among the 403 participants who had at least one positive bacteriological result at baseline, 168 (41.7%) were recorded as cured; of these, 83/205 (39.9%) were in the control arm compared with 85/195 (43.6%) in the intervention arm. The OR of being cured was 1.16 (95% CI

0.83 to 1.63) in the intervention vs the control arm and was similar to the adjusted OR (tables 3 and 4).

#### Continuous smoking abstinence
Among those who identified as cigarette smokers at baseline (345 (60.1%)), 27 had information (self-report plus biochemical verification) to enable the identification of continuous abstinence at 6 months, of which 22 had continuously abstained from smoking. These were similarly distributed across the two study arms: 10 (5.59%) participants in the intervention arm compared with 12 (7.23%) in the control arm (OR 0.76, 95% CI 0.35 to 1.63) (tables 3 and 4). At the 3-month follow-up, 20 (11.2%) participants in the intervention arm compared with 27 (16.3%) in the control arm continuously abstained from smoking (OR 0.65, 95% CI 0.37 to 1.14) (tables 3 and 5).

**Table 2** Baseline descriptive alcohol and smoking characteristics by study arm

| | Control (N=291) n (%)* | Intervention (N=283) n (%)* | Total (N=574) n (%)* |
|---|---|---|---|
| In the past month, smoked tobacco | | | |
| Not at all† | 110 (37.8) | 92 (32.5) | 202 (35.2) |
| Daily | 149 (51.2) | 149 (52.7) | 298 (51.9) |
| Less than daily | 32 (11.0) | 42 (14.8) | 74 (12.9) |
| Had a drink in the past 12 months | 208 (71.5) | 223 (78.8) | 431 (75.1) |
| AUDIT score (men) : mean (SD) (max: 19)‡ | 12.27 (3.98) | 13.02 (3.78) | 12.66 (3.89) |
| AUDIT score (women): mean (SD) (max: 19)‡ | 11.32 (4.02) | 10.98 (4.02) | 11.15 (4.0) |
| Hazardous/harmful drinking and smoking combined (constructed) | | | |
| Hazardous/harmful drinking only§ | 110 (37.8) | 92 (32.5) | 202 (35.2) |
| Smoking only | 103 (35.4) | 77 (27.2) | 180 (31.4) |
| Smoking and hazardous/harmful drinking§ | 78 (26.8) | 114 (40.3) | 192 (33.4) |

*Frequencies and (percentages) are presented unless otherwise stated.

†Non-smokers were included only if they were harmful or hazardous drinkers.

‡Only hazardous/harmful drinkers and/or current smokers were included in the study. Therefore, patients with TB were excluded if they were non-current smokers and had an AUDIT score of <7 (women) or <8 (men) or 19; however, they were included if they were smokers independent of whether they had a drink in the past year and therefore independent of the AUDIT score. These AUDIT scores are thus representative of the mean AUDIT scores in the entire study sample and differ from the AUDIT score in the harmful/hazardous drinkers whose change in AUDIT score was measured at 3 and 6 months of follow-up.

§Harmful/hazardous drinking is defined as having an AUDIT scores of ≥8 for men or ≥7 for women but <20.

AUDIT, Alcohol Use Disorders Identification Test.

### Change in harmful/hazardous drinking

AUDIT scores were about four points lower at both follow-up times than at baseline, independent of the intervention (table 3). In the intervention arm, participants had, on average, a reduction of 0.04 points (95% CI −2.0 to 1.91) on the AUDIT score at 6 months, compared with those in the control arm controlling for baseline scores, whereas an average increase of 0.55 (95% CI −1.01 to 2.11) was observed at 3 months (tables 4 and 5).

### Medication adherence and ART uptake

At 6 months, the OR of taking ART medication was 2.05 (95% CI 0.80 to 5.27) comparing the intervention arm to the control arm and controlling for ART baseline medication status, whereas it was 0.79 (95% CI 0.38 to 1.65) at 3 months. The proportion of participants who had optimal TB medication adherence was 90.2% (120/133) at 6 months and 91.7% (319/348) at 3 months. Suboptimal TB medication adherence ORs were 0.89 (95% CI 0.26 to 3.07) and 1.22 (95% CI 0.52 to 2.87) comparing intervention arm to the control arm at 6 and 3 months, respectively. The proportions of participants on ART who had optimal ART medication adherence were high at both 3 months (165/167, 98.8%) and 6 months (139/143, 97.2%) of follow-up. Suboptimal ART medication adherence ORs were 1.17 (95% CI 0.14 to 9.94) and 1.58 (95% CI 0.10 to 26.12) comparing the intervention arm to the control arm at 6 and 3 months, respectively. (tables 3–5)

### Intervention fidelity
#### MI fidelity

The recordings of 17 counsellors (one each) were transcribed verbatim and then assessed. In terms of the global ratings, the LHWs' counselling sessions were above proficiency levels on all items, namely, cultivating change talk, softening sustain talk, partnership and empathy (as the mean scores were all above 2). In terms of the summary measures, the LHWs' counselling sessions did not achieve the basic proficiency threshold of 3.5 for the relational component (partnership+empathy) as their mean score was 3.1 (SD 1.19). However, their mean score on the technical component (cultivating change talk+softening sustain talk) of 3.3 (SD 0.97) was above the threshold of 3. For behavioural counts, 'asking questions' had the highest mean score (24.2, SD 10.42), followed by 'affirm', with a mean score of 5.5 (SD 3.7). The counsellors were least likely to engage in the following: persuade with permission and emphasising autonomy. The mean reflections to questions ratio was 0.23 (SD 0.24). The LHWs made on average 9.3 (SD=4.74) MI adherent (affirm, emphasise autonomy and seek collaboration) and 1.2 (SD 2.28) MI non-adherent (confront and persuade) statements per session (online supplemental table 4).

#### SMS delivery

Of the total number of information–motivation–behaviour messages triggered, 3583 (80.4%) were delivered. All due SMS messages were delivered to 95 (41.9%)

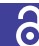

**Table 3** Descriptive statistics for primary and secondary outcomes by study arm at baseline (where available), 3 months (where available) and 6 months

| | Baseline n (%)* | | | 3 months of follow-up n (%)* | | | 6 months of follow-up n (%)* | | |
|---|---|---|---|---|---|---|---|---|---|
| | Control | Intervention | Total | Control | Intervention | Total | Control | Intervention | Total |
| **TB treatment status†** | | | | | | | | | |
| Successful‡ | | | | | | | 204 (70.1) | 192 (67.8) | 396 (69.0) |
| Not Successful | | | | | | | 87 (29.9) | 91 (32.16) | 178 (31.01) |
| **Cured†,§** | | | | | | | | | |
| Yes | | | | | | | 83 (39.9) | 85 (43.6) | 168 (41.7) |
| No | | | | | | | 125 (60.1) | 110 (56.4) | 235 (58.3) |
| **Continuous smoking abstinence¶** | | | | | | | | | |
| Yes | | | | 27 (16.3) | 20 (11.2) | 47 (13.6) | 12 (7.2) | 10 (5.6) | 22 (6.4) |
| No | | | | 139 (83.7) | 159 (88.8) | 298 (86.4) | 154 (92.8) | 169 (94.4) | 323 (93.6) |
| Harmful/hazardous drinkers** (N) | 188†† | 206†† | 394†† | 141†† | 130†† | 271†† | 112†† | 127†† | 239†† |
| AUDIT score, mean (SD) | 12.76 (3.42) | 13.12 (3.47) | 12.94 (3.45) | 8.28 (6.18) | 8.84 (5.38) | 8.55 (5.81) | 8.79 (6.66) | 8.70 (5.83) | 8.74 (6.22) |
| Difference from baseline, mean (SD) | | | | −4.61 (6.26) | −4.07 (5.33) | −4.35 (5.83) | −4.25 (6.56) | −4.17 (6.61) | −4.21 (6.57) |
| HIV-positive patients | 163†† (56.2) | 142†† (50.2) | 305†† (53.2) | 122†† | 83†† | 205†† | 100†† | 83†† | 183†† |
| Taking ART medication if HIV positive‡‡ | 139 (85.3) | 115 (81.0) | 254 (83.3) | 91 (74.6) | 58 (69.9) | 149 (72.7) | 80 (80.0) | 74 (89.2) | 154 (84.2) |
| **ART medication adherence¶** | | | | | | | | | |
| Optimal adherence | | | | 101 (99.0) | 64 (98.5) | 165 (98.8) | 75 (97.4) | 64 (97.0) | 139 (97.2) |
| Suboptimal adherence | | | | 1 (1.0) | 1 (1.54) | 2 (1.2) | 2 (2.6) | 2 (3.0) | 4 (2.8) |
| **TB medication adherence** | | | | | | | | | |
| Optimal adherence | | | | 181 (92.3) | 138 (90.8) | 319 (91.7) | 61 (89.7) | 59 (90.8) | 120 (90.2) |
| Suboptimal adherence | | | | 15 (7.6) | 14 (9.2) | 29 (8.3) | 7 (10.3) | 6 (9.2) | 13 (9.8) |

*Frequencies and (percentages) are presented unless otherwise stated.
†Only assessed at 6-months.
‡Primary outcome: this is a binary variable defined as either successful treatment (cured or treatment completed) or failed treatment, death, acquired drug resistance, loss to follow-up or treatment interrupted for more than 2 months, or outcome not evaluated/unknown.
§Based on having a cured treatment outcome among those who were bacteriologically positive at baseline.
¶Assessed at 3 and 6 months; this table refers to *cigarette* smokers only (other forms of tobacco smoking are excluded).
**Hazardous/harmful drinkers who are not alcohol dependent=AUDIT score of ≥8 for men or ≥7 for women but <20; *|* Important distinction at baseline for eligibility purposes.
††Denominator for the mean (SD) or denominator for %.
‡‡Information on HIV positivity was obtained from information from TB records combined with patient self-report at baseline. True HIV-positivity rates may have been higher.
ART, antiretroviral therapy; AUDIT, Alcohol Use Disorders Identification Test; TB, tuberculosis.

**Table 4** Regression analysis results for the primary and secondary outcomes at 6 months

| | Crude OR (95% CI)* | P value* | Adjusted OR (95% CI)* | P value* |
|---|---|---|---|---|
| **Primary outcome** | | | | |
| TB treatment status: successful (ref: not successful) | 0.90 (0.64 to 1.27) | 0.548 | 0.86† (0.60 to 1.24) | 0.421 |
| **Secondary outcomes** | | | | |
| Cured (ref: not cured) | 1.16 (0.83 to 1.63) | 0.374 | 1.07† (0.76 to 1.51) | 0.684 |
| Continuous smoking abstinence (ref: no)‡ | 0.76 (0.35 to 1.63) | 0.482 | | |
| TB medication adherence (ref: optimal) | 0.89 (0.26 to 3.07) | 0.849 | | |
| ART medication adherence (ref: optimal) | 1.17 (0.14 to 9.94) | 0.884 | | |
| Taking ART medication (ref: no) | 2.05§ (0.80 to 5.27) | 0.136 | | |
| AUDIT | −0.04¶ (−2 to 1.91) | 0.966 | 0.02** (−1.55 to 1.6) | 0.976 |

*Analyses accounted for potential clustering by centre.
†Adjusted for district, sex, and smoking/drinking status and HIV status at baseline. It is worth noting that of the variables in the adjusted model, the only statistically significant result is for the district variable.
‡Given the limited number of those who were identified as continually abstained, we were only able to adjust for one additional variable at a time. Adding one of the following variables: heaviness of smoking, type of drinker at baseline, age when started smoking and the duration of smoking at baseline, the adjusted OR of continuous abstinence comparing the intervention to the control arm ranged between 0.73 and 0.76 with similar confidence limits as for the crude estimate.
§Adjusting for ART status at baseline.
¶Controlling for the AUDIT baseline values; the values represent the study arm regression coefficient.
**Controlling for the AUDIT baseline values and adjusted for district, sex, and smoking/drinking status and HIV status at baseline; the values represent the study arm regression coefficient.
ART, antiretroviral therapy; AUDIT, Alcohol Use Disorders Identification Test; TB, tuberculosis.

of the participants who completed the first MI (see online supplemental table 5 for more details).

## Costs and health-related quality of life
Unit costs used to estimate the mean costs are presented in online supplemental table 6. Incremental cost:utility ratios are not presented since the intervention was not clinically effective. The mean cost of the ProLife intervention was ZAR 2601 (SD 6) ($180.02, SD $0.42) per participant in the intervention arm (n=283). The mean cost of usual care was ZAR 681 (SD 357) ($47.13, SD $24.71)

in the intervention arm (n=122) vs ZAR 706 (SD 302) ($48.86, SD $20.90) in the control arm (n=131). The total mean cost of care including the intervention was ZAR 3285 (SD 357) ($227.37, SD 24.71) in the intervention arm (n=122). EQ-5D-3L data were available at the three time points for 137 intervention and 159 control arm participants. The mean QALYs estimated over 6 months were 0.442 (SD 0.061) in the intervention arm vs 0.430 (SD 0.074) in the control arm (adjusted mean difference 0.006, 95% CI −0.001 to 0.013).

**Table 5** Regression analysis results for secondary outcomes measured at 3 months

| Secondary outcome | Crude OR (95% CI)* | P value* | Adjusted OR (95% CI)* | P value* |
|---|---|---|---|---|
| Continuous smoking abstinence (ref: no)† | 0.65 (0.37 to 1.14) | 0.135 | | |
| TB medication adherence (ref: optimal) | 1.22 (0.52 to 2.87) | 0.641 | | |
| ART medication adherence (ref: optimal) | 1.58 (0.10 to 26.12) | 0.750 | | |
| Taking ART medication (ref: no) | 0.79‡ (0.38 to 1.65) | 0.53 | 0.74§ (0.35 to 1.58) | 0.443 |
| AUDIT | 0.55¶ (−1.01 to 2.11) | 0.474 | 0.74** (−0.62 to 2.1) | 0.273 |

*Analyses accounted for clustering.
†Given the limited number of those who were identified as continually abstained, we were only able to adjust for one additional variable at a time. Adding one of the following variables: heaviness of smoking, type of drinker at baseline, age when started smoking and the duration of smoking at baseline, the adjusted OR of continuous abstinence comparing the intervention to the control arm ranged between 0.63 and 0.66 with similar confidence limits as for the crude estimate.
‡Adjusting for art status at baseline.
§Adjusted for art status at baseline, district, sex, and smoking/drinking status and HIV status at baseline.
¶Controlling for the AUDIT baseline values; the values represent the study arm regression coefficient.
**Controlling for the AUDIT baseline values and adjusted for district, sex, and smoking/drinking status and HIV status at baseline; the values represent the study arm regression coefficient.
ART, antiretroviral therapy; AUDIT, Alcohol Use Disorders Identification Test; TB, tuberculosis.



## DISCUSSION

This RCT did not provide evidence for improved TB treatment success rates in those receiving the ProLife intervention compared with those receiving usual care. We could also not demonstrate significant beneficial effects on any of the secondary outcomes, that is, smoking, alcohol consumption, medication adherence and ART initiation. To our knowledge, there are no other published studies of similar complex interventions that aim to improve TB treatment outcomes in patients who smoke or drink to harmful or hazardous extent. Interventions evaluated by other studies were either complex interventions or SMS-based interventions aimed at improving TB outcomes through the pathway of increasing adherence, but without an alcohol or smoking intervention component[44][45] or focused on a single behaviour, namely, smoking or drinking.[46][47] Of the latter studies, a brief smoking cessation intervention was effective in inducing smoking cessation in patients with TB but did not improve TB outcomes.[46] Conversely, in another study in India, intensive counselling for alcohol disorders led to significantly better TB treatment outcomes in the intervention group compared with the control group.[47] Smoking cessation also led to better TB treatment outcomes in a secondary analysis of a large tobacco cessation trial in patients with TB in Bangladesh and Pakistan.[48] Our non-significant result for smoking-related outcomes is not consistent with findings from our previous TB study, which used a single MI session and found that the chance of sustained smoking cessation was twice as high in the MI intervention group compared with the control group,[14] although with a less stringent exhaled CO cut-off point. Evidence on the effectiveness of MI for smoking abstinence in non-TB settings has been equivocal.[49] Self-reported alcohol consumption decreased with about 4 points in both intervention and control arms in our study at both follow-up times. Answering questions on drinking in brief intervention trials may alter subsequent self-reported behaviour: exposing non-intervention control groups to an integral component of the intervention may therefore underestimate the effect of the intervention.[50] There have been few previous studies looking at MI and SMS interventions for the modification of hazardous/harmful drinking in the context of TB. A previous trial of a brief counselling intervention to reduce alcohol consumption in patients with TB did not find a significant effect on alcohol reduction.[51] Outside a TB setting, results have been mixed. A meta-analysis showed a small but significant improvement in outcomes when MI was used in conjunction with cognitive behavioural therapy for comorbid alcohol use and depression.[52] Self-reported TB and ART medication adherence was high overall in our study population, which is consistent with other studies conducted in South Africa.[53][54] It is possible that we did not find a difference in treatment adherence due to a ceiling effect.

There were several key strengths in this RCT. This was an individual RCT with a relatively large sample size and a high follow-up rate (87%) for the primary outcome.

Primary outcome assessment was blinded. This was a novel intervention, which built on previous successes with both MI and mHealth interventions and was aligned with the WHO's call to increase the use of digital technologies to improve TB care.[15] We used a validated alcohol consumption questionnaire (AUDIT)[29] and a 4-day timeline follow-back for medication adherence to reduce recall bias as self-reports tend to under-report drinking while overestimating adherence behaviour.[35][55] Smoking cessation was confirmed with exhaled CO using strict cut-off points. Overall, the quality of the counselling was acceptable. The results of our MI analyses suggest that the LHWs trained as counsellors were more proficient in MI than during the feasibility stage, as observed by their global rating scores on cultivating change talk, softening sustain talk, partnership and empathy (online supplemental table 4). These results were achieved by ongoing monitoring and training of LHWs during the trial and adapting the training based on feedback from the feasibility stage. Extra counsellors were also appointed to minimise travel distances to clinics. There were some limitations associated with this RCT. Trial recruitment had to be terminated before the planned sample size because of funding and time constraints. Nevertheless, the calculation of sample size was based on an anticipated 25% Loss To Follow Up for the primary outcome, while in reality, only 13.4 %, of the TB outcomes were not available. As a result, we achieved a slightly higher power to detect the a 10% difference in primary outcomes than that we had aimed for (83% vs 80%). The smaller sample size did, however, reduce the power to detect a difference for secondary outcomes for which the LTFU was much higher than 25%. Also, the calculated sample size was not powered for subgroup analysis, which was the case for outcomes relating to smoking, drinking, ART and cure rates. In addition, due to the COVID-19 lockdown in March 2020, we had to switch to telephonic follow-up of participants using a shortened questionnaire (22 participants) and could not access clinics to retrieve outstanding TB treatment outcomes. The low intervention uptake meant that half of the participants received only one or two MI sessions combined with SMS messages. SMS messages were only used for the first half of the study period, and one-quarter of participants did not receive their messages, a commonly occurring problem in LMICs.[20][56] It could be argued that in the absence of ongoing text messages, the MI and associated text messages were not enough to keep participants focused for the second 3 months of the trial. The two-arm study design did not permit the untangling of the individual effects of SMS and MI. Understanding their separate effects could have important cost implications as SMS communication would be cheaper and easier to organise than individual counselling.

The lack of effectiveness of our intervention on the primary outcome (TB treatment success) can have a number of possible explanations. Although intervention uptake was high (80.2%) for the first counselling session, many participants did not return for the second

(29.7%) and third (47%) sessions. As a result of this, only about half of the intervention arm participants received all three MI sessions. Furthermore, about one-quarter of all participants did not receive any SMS messages. Low intervention uptake leads to a dilution of any potential effects. The lack of effectiveness on TB treatment success could perhaps also be explained by the complexity of the ProLife intervention itself: counsellors had to address multiple behaviours, namely, medication adherence, tobacco smoking and hazardous/harmful drinking. Despite having established the feasibility and acceptability of this approach[31] and ongoing on-site performance monitoring and feedback of counsellors, it is possible that MI for multiple behaviour change in the ProLife study was counterproductive as counsellors may have ended up not focusing on any of the behaviours at optimal levels. Similarly, patients might have found it difficult to change multiple behaviours simultaneously, especially because smoking and drinking are mutually reinforcing. This integrated approach was nevertheless adopted to avoid the need for multiple vertical counselling services (in addition to TB treatment and HIV treatment), to allow the different elements of the programme to reinforce one another, and to improve the affordability, feasibility and acceptability for a future roll-out of the programme. It is also possible that sequential interventions may be better, at least for smoking cessation.[57] More intensive counselling (more sessions) or a modified counselling method may have been more appropriate, even more so since a recent review of reviews of MI casts doubt on its efficacy.[49 58 59] For example, more emphasis on increasing patient knowledge, in addition to increasing self-efficacy, may have been more effective.[60] The cause of the mhealth message delivery problems (such as poor network coverage and no electricity to charge phones)[61] would need to be investigated in order to increase the effectiveness of future mHealth interventions. Messages may also have to be intensified or modified to be more interactive and/or tailored to specific circumstances of each individual. This would improve the personal value of the intervention to the individual, which is likely to increase the chances of their participation in the intervention.[62] Consistent with the normalisation process theory,[63] cognitive participation in the intervention might have been higher had we been deliberate in the implementation to ensure the TB nurse, who would have routinely seen the participants, provided additional support and motivated participants to attend MI sessions with the counsellor. In this way, the intervention would have gained 'legitimacy', but this would have led to unblinding of the nurses to the intervention arm.

In conclusion, we could not demonstrate that the ProLife intervention was effective in improving TB treatment outcomes. This may be due to the lack of effect of the intervention, but the study may also have been underpowered for the intermediary secondary outcomes. Valuable lessons were learnt on challenges relating to training LHWs in MI counselling and delivery, SMS delivery in a challenging socioeconomic context and the reasons for loss to follow-up of TB participants with multiple health problems. Further research is needed to provide answers on how to increase intervention uptake in poor resource settings and whether our complex intervention should have been more intensive. Other important questions are whether another counselling method would have been more effective. Lastly, in the light of the already existing evidence of SMS and the costs and implementation challenges relating to MI, intervention studies limited to an mHealth intervention but using different intensities, duration and type of interventions (one-way, two-way and interactive) are needed.

**Author affiliations**
[1]Research, Postgraduate Studies and Innovation, Sefako Makgatho Health Sciences University, Pretoria, South Africa
[2]School of Health Systems and Public Health, University of Pretoria, Pretoria, South Africa
[3]Department of Health Sciences, University of York, York, UK
[4]Department of Psychology, University of Johannesburg, Auckland Park, South Africa
[5]Alcohol, Tobacco and Other Drug Research Unit, Medical Research Council of South Africa, Pretoria, South Africa
[6]Department of Health Sciences and the Hull York Medical School, University of York, York, UK
[7]Department of Family Medicine, University of the Witwatersrand, Johannesburg-Braamfontein, South Africa
[8]Department of Family Medicine, Sefako Makgatho Health Sciences University, Pretoria, South Africa
[9]Norwich Medical School, University of East Anglia, Norwich, UK
[10]Africa Centre for Tobacco Industry Monitoring and Policy Research, Sefako Makgatho Health Sciences University, Pretoria, South Africa

**Acknowledgements** We are grateful to the following people: Ms M Malefo for her work as project manager, Ms D Bell for conducting the motivational interviewing training, Ms N Kitleli for supervising the lay counsellors, Mesadie Machaea-Malinga for conducting the fidelity assessments, the district and clinic TB managers and TB nurses, the fieldworkers and the lay counsellors. The authors are also grateful to the National Department of Health and the three Provincial Departments of Health for granting permission for this study.

**Contributors** GL led the study design and manuscript preparation and contributed to the monitoring of the study implementation. MK contributed to the study design and the manuscript, led the sample size determination and did the statistical analysis. NKM contributed to study design, particularly with respect to development of the motivational interviewing (MI) intervention, did the analysis of the MI data and contributed to the manuscript writing. AVZ led the data management and short message service system, monitored data quality and contributed to data analysis and manuscript writing. ASM contributed to study design and led on the preparation of the manuscript. NDM contributed to the study design, intervention development and manuscript preparation. JL and SP led the economic evaluation. AT contributed to the economic evaluation. JT was site lead for the Bojanala subdistrict. OBO was site lead for the Sedibeng subdistrict. MB contributed to the study design, data interpretation and the manuscript. KS and OAA-Y are coprincipal investigators. Both contributed to study design, data interpretation and manuscript writing. All authors read and approved the final draft of the manuscript. OAA-Y is the guarantor.

**Funding** This project was funded by the SA-Medical Research/Newton Foundation Grant on TB control implementation science (UK/South Africa Newton Fund RFA: TB control implementation science (MRC-RFA-02: TB -05-2015)). This UK funded award is part of the EDCTP2 programme supported by the European Union.

**Competing interests** OAA-Y, the principal investigator, received a research grant (MRC-RFA-02: TB -05-2015) which was allocated to Sefako Makgatho Health Sciences University to pay for the expenses related to the research project, and received support for attending meetings and for travel by Tax Justice Network Africa. He is the vice chair of the Standing Committee on Health, Academy of Science, SA (not remunerated). As coprincipal investigator, KS received part

of the MRC-Newton research grant (MRC-RFA-02: TB -05-2015), which was allocated to the University of York to support the research by the authors affiliated to the University of York (KS, NDM, MK, SP and JL). GL received a small monthly remuneration per month from 1 October 2018 to 30 June 2020 as academic research coordinator. AVZ was employed as a research data manager on a part-time basis from 1 September 2017 to 30 June 2020.

**Patient consent for publication** Not applicable.

**Ethics approval** This study involves human participants and was approved by the research ethics committees of Sefako Makgatho Health Sciences University (SMU) (ref: SMUREC/D/234/2017: IR); the South African Medical Research Council (with the SMU Ethics Committee serving as the Research Ethics Committee Record); the University of Pretoria (Ref: 434/2017); the University of Witwatersrand (M160455); and the University of York (no reference number, approval date 15 January 2017 (29). Participants gave informed consent to participate in the study before taking part. The trial protocol was previously published.

**Provenance and peer review** Not commissioned; externally peer reviewed.

**Data availability statement** Data are available in a public, open access repository. The study protocol was previously published (ImPROving TB Outcomes by Modifying LIFE-style Behaviours through a Brief Motivational Intervention Followed by Short Text Messages (ProLife): study protocol for a randomised controlled trial). The deidentified participant and SMS data sets are stored in labelled Stata files and are accompanied by a statistical analysis plan and metadata explaining each variable. Data will be embargoed for data analysis until 30 June 2023. Thereafter, permission must be obtained from the principal investigators (OAA-Y and KS) for any data analysis not yet performed by the primary research group. Data are stored in the institutional data repository at Sefako Makgatho Health Sciences University called Discover research (https://smu-za.figshare.com/) with a CC-BY 4.0 (Attribution) license (Creative Commons — Attribution 4.0 International—CC BY 4.0). The statistical analysis plan is available as supplementary material.

**Open access** This is an open access article distributed in accordance with the Creative Commons Attribution 4.0 Unported (CC BY 4.0) license, which permits others to copy, redistribute, remix, transform and build upon this work for any purpose, provided the original work is properly cited, a link to the licence is given, and indication of whether changes were made. See: https://creativecommons.org/licenses/by/4.0/.

**ORCID iDs**
Goedele Louwagie http://orcid.org/0000-0002-4384-2318
Andre Van Zyl http://orcid.org/0000-0001-6872-5276
Andrew Stephen Moriarty http://orcid.org/0000-0003-0770-3262
Jinshuo Li http://orcid.org/0000-0003-1496-7450
Kamran Siddiqi http://orcid.org/0000-0003-1529-7778
Max Bachmann http://orcid.org/0000-0003-1770-3506

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
