## [Reviewer comments · BMJ Open]

ARTICLE DETAILS

TITLE (PROVISIONAL)	Effect of a brief motivational interview and text message intervention targeting tobacco smoking, alcohol use, and medication adherence to improve tuberculosis treatment outcomes in adult patients with tuberculosis: a multicentre, randomised controlled trial of the ProLife programme in South Africa
AUTHORS	Louwagie, Goedele; Kanaan, Mona; Morojele, Neo; Van Zyl, Andre; Moriarty, Andrew; Li, Jinshuo; Siddiqi, Kamran; Turner, Astrid; Mdege, Noreen; Omole, Olufemi; Tumbo, John; Bachmann, Max; Parrott, Steve; Ayo-Yusuf, Olalekan

VERSION 1 – REVIEW

REVIEWER	Mars, Maurice University of Kwazulu-Natal, TeleHealth
REVIEW RETURNED	21-Sep-2021

GENERAL COMMENTS	Thank you for the opportunity to review this paper reporting a randomised controlled study. The study investigates the effect on treatment outcome of the combined use of motivational interventions provided by lay health workers and supporting SMS messages sent to patients with drug sensitive tuberculosis. Secondary evaluations of treatment adherence and reduction in smoking and alcohol use and cost analysis are provided. This study shows no demonstrable benefit of the intervention. There are a number of possible reasons for this, not all of which are identified by the authors. Attendance at motivational interventions was poor, so in effect, this can be considered as another study on the use of SMS messages to improve medication adherence. It is different to many other studies as SMS messages were only used for the first half of the study period, and one quarter of participants did not receive their messages. It could also be argued that in the absence of ongoing text messages, the motivational interventions and associated text messages were enough to keep participants focussed for the second three months of the trial. This cannot be confirmed as there are no comparative data presented, excluding those who dropped out at the different phases, or data specific to those who completed the trial. For example, how many motivational interventions and SMS messages did those who completed the study not receive? The sample was 28% less than required. What is the effect of this on the results?
--

	Many references are dated and appear to reflect those used in the submission of the protocol; only 10 of 58 references are recent. Have there been any significant changes in the reported literature since 2017? For example, in lines 114-118, the references in support of the benefits of mHealth and SMS use are from 2013 and 2017. An interesting observation is that some subjects did not receive their SMS messages. The explanation seems to be repetitive. What is the difference between being 'disconnected from the network', having a number that is deactivated, or a phone that is off for a long time? Phones that are off for a long time still receive the SMS eventually. Discussion, para 2, lines 9-11: the switch to telephonic follow-up should be reported in the Methods. The study protocol was published, but with hindsight, was the study design appropriate? Having had a 'no effect' outcome, it would be helpful to identify possible weaknesses and shortcomings of the study design. The findings could be contextualised within the specific literature, both recent and past, on studies that did not show an effect of either motivational interventions and SMS messages in this patient population.
--	---

REVIEWER	Shibu, Vijayan
REVIEW RETURNED	24-Sep-2021

GENERAL COMMENTS	Appreciate the attempt to publish though the outcomes are not positive. This also reinforces that fact that habits and substance abuse need to be addressed as a disease or more rigorous interventions needed . Soft touch interventions may fall short.
---

REVIEWER	Gupte, Himanshu Narotam Sekhsaria Foundation
REVIEW RETURNED	01-Oct-2021

GENERAL COMMENTS	Congratulations for a well written article especially when the results were not favourable! Overall, it is a well written and well-explained article giving an insight into the complex intervention and challenges faced in its implementation. While most aspects have been logically discussed in detail, I find some lack of clarity on the implementers' perspective. The LHWs who were the most important aspect of the intervention have been discussed superficially. This was a first trial with a complex intervention involving MI by LHWs. The discussion on probable reasons of the outcomes does not include their role. While there is discussion about whether the intervention should have been more intensive or some other technique should have been used, the assessment of the LHWs was based on only 20 minutes of interview. Is this sufficient to assess this? Further, there were some red flags in the feasibility study about failure to reach MI threshold levels by LHWs and some implementation challenges were also mentioned by them. How was this addressed in the intervention? Another question is that when the intended intervention could not fully implemented due to various reasons discussed, is it appropriate to say that it was not effective?
--

	The article is worth publishing for dissemination of these interesting findings and learnings but some more light has to be shed on these aspects pointed out here.
--	---

VERSION 1 – AUTHOR RESPONSE

Reviewer: 1

Prof. Maurice Mars, University of Kwazulu-Natal

Comments to the Author:

Thank you for the opportunity to review this paper reporting a randomised controlled study. The study investigates the effect on treatment outcome of the combined use of motivational interventions provided by lay health workers and supporting SMS messages sent to patients with drug sensitive tuberculosis. Secondary evaluations of treatment adherence and reduction in smoking and alcohol use and cost analysis are provided.

This study shows no demonstrable benefit of the intervention. There are a number of possible reasons for this, not all of which are identified by the authors. Attendance at motivational interventions was poor, so in effect, this can be considered as another study on the use of SMS messages to improve medication adherence. It is different to many other studies as SMS messages were only used for the first half of the study period, and one quarter of participants did not receive their messages. It could also be argued that in the absence of ongoing text messages, the motivational interventions and associated text messages were enough to keep participants focussed for the second three months of the trial. This cannot be confirmed as there are no comparative data presented, excluding those who dropped out at the different phases, or data specific to those who completed the trial. For example, how many motivational interventions and SMS messages did those who completed the study not receive?

Thank you very much for your valuable feedback. Note that SMS-messages were only initiated after the completion of the first MI. Therefore, participants who received SMS had received at least one MI counselling session (with the exception of 1 participant, see Supplementary table)

We added the following to the limitations' paragraph of the manuscript (p23):

“The intervention was therefore limited to one or two MI sessions combined with SMS messages in nearly half of the participants. Furthermore, SMS messages were only used for the first half of the study period, and one quarter of participants did not receive their messages. It could be argued that in the absence of ongoing text messages, the motivational interventions and associated text messages were not enough to keep participants focussed for the second three months of the trial.”

The sample was 28% less than required. What is the effect of this on the results?

Thank you. This should not have impacted on our anticipated power to detect a difference for the primary outcome for which the LTFU was only 13.4 %, much lower than the anticipated 25%. If we calculate the power to detect a difference of 10% (proportions of 0.86 in the intervention vs 0.76 in the control group) and use our actual achieved sample size of 283 in the intervention group, then the calculated power to detect a difference is 0.83. This is in fact slightly higher than our desired power of 0.80. Our conclusions regarding the primary outcome are therefore not influenced by the lower than planned sample size. Also note: we already knew that we would achieve a high proportion of the primary outcomes when we decided to stop the enrolment earlier. This assisted us in making the difficult decision which was necessitated by funding constraints.

However, it is certainly correct that the smaller final sample size impacted the power to detect a difference for secondary outcomes where the LTFU was much higher than the anticipated 25%.

It is also important to note that the sample size was not powered for these secondary outcomes which – with the exception of adherence information-only applied to subgroups of drinkers or smokers or HIV-positive people or patients diagnosed based on positive bacteriology.

The following was added to the limitations of the study:

“Trial recruitment had to be terminated before the planned sample size because of severe funding and time constraints. Nevertheless the calculation of sample size was based on an anticipated 25% LTFU for the primary outcome, while in reality only 13.4 %, of the TB outcomes were not available. As a result, we achieved a slightly higher power to detect the a 10% difference in primary outcomes, than we had aimed for (83% vs. 80%) . The smaller sample size did however reduce the power to detect a difference for secondary outcomes for which the LTFU was much higher than 25%. Furthermore the calculated sample size was not powered for subgroups analysis which was the case for outcomes relating to smoking, drinking, ART and cure rates.”

Many references are dated and appear to reflect those used in the submission of the protocol; only 10 of 58 references are recent. Have there been any significant changes in the reported literature since 2017? For example, in lines 114-118, the references in support of the benefits of mHealth and SMS use are from 2013 and 2017.

We updated the references where possible and appropriate. The list below are references replacing older references or have been added anew.

Thomas BE, Thiruvengadam K, Rani S, et al. Smoking, alcohol use disorder and tuberculosis treatment outcomes: A dual co-morbidity burden that cannot be ignored. *PLoS ONE* 14(7): e0220507. <https://doi.org/10.1371/journal.pone.0220507> [Accessed 4 Nov 2021].

Imtiaz S, Shield KD, Roerecke M, et al. Alcohol consumption as a risk factor for tuberculosis: meta-analyses and burden of disease. *Eur Respir J* 2017; 50: 1700216. [<https://doi.org/10.1183/13993003.00216-2017>].

Ogbo FA, Ogeleka P, Okoro A, et al. Tuberculosis disease burden and attributable risk factors in Nigeria, 1990–2016. *Tropical Medicine and Health* (2018) 46:34. <https://doi.org/10.1186/s41182-018-0114-9>.

Azeez A, Mutambayi R, Odeyemi A, et al. Survival model analysis of tuberculosis treatment among patients with human immunodeficiency virus coinfection. *Int J Mycobacteriol* 2019;8:244-51.

Mollel EW, Chilongola JO. Predictors for Mortality among Multidrug-Resistant Tuberculosis Patients in Tanzania. *Journal of Tropical Medicine* 2017. <https://doi.org/10.1155/2017/9241238>

Harder VS, Musau AM, Musyimi CW, Ndetei DM, Mutiso VN. A randomized clinical trial of mobile phone motivational interviewing for alcohol use problems in Kenya. *Addiction*. 2020 Jun;115(6):1050-1060. doi: 10.1111/add.14903. Epub 2020 Jan 3. PMID: 31782966; PMCID: PMC8353663.

Ragan EJ, Kleinman MB, Sweigart B, Gnatienko N, Parry CD, Horsburgh CR, LaValley MP, Myers B JK. The impact of alcohol use on tuberculosis treatment outcomes: a systematic review and meta-analysis. *Int J Tuberc Lung Dis*. 2020;24(1):73–82.

Shah R, Watson J, Free C. A systematic review and meta-analysis in the effectiveness of mobile phone interventions used to improve adherence to antiretroviral therapy in HIV infection. *BMC Public Health*. 2019 Jul 9;19(1):915.

Palacio A, Garay D, Langer B, et al. Motivational Interviewing Improves Medication Adherence: a Systematic Review and Meta-analysis. *Gen Intern Med.* 31(8):929–40. DOI: 10.1007/s11606-016-3685-3.

Madhombiro M, Kidd M, Dube B, Dube M, Mutsvuke W, Muronzie T, Zhou DT, Derveeuw S, Chibanda D, Chingono A, Rusakaniko S, Hutson A, Morse GD, Abas MA, Seedat S. Effectiveness of a psychological intervention delivered by general nurses for alcohol use disorders in people living with HIV in Zimbabwe: a cluster randomized controlled trial. *J Int AIDS Soc.* 2020 Dec;23(12):e25641. doi: 10.1002/jia2.25641. PMID: 33314786; PMCID: PMC7733606.

Gashu KD, Gelaye KA, Mekonnen ZA, Lester R, Tilahun B. Does phone messaging improve tuberculosis treatment success? A systematic review and meta-analysis. *BMC Infect Dis.* 2020;20(1):42. Published 2020 Jan 14. doi:10.1186/s12879-020-4765-x

Ngwatu, B. K., Nsengiyumva, N. P., Oxlade, O., Mappin-Kasirer, B., Nguyen, N. L., Jaramillo, E., Falzon, D., Schwartzman, K., & Collaborative group on the impact of digital technologies on TB (2018). The impact of digital health technologies on tuberculosis treatment: a systematic review. *The European respiratory journal*, 51(1), 1701596. <https://doi.org/10.1183/13993003.01596-2017>

Moosa, A., Gengiah, T.N., Lewis, L. et al. Long-term adherence to antiretroviral therapy in a South African adult patient cohort: a retrospective study. *BMC Infect Dis* 19, 775 (2019). <https://doi.org/10.1186/s12879-019-4410-8>

Webb Mazinyo E, Kim L, Masuku S, Lancaster JL, Odendaal R, et al. (2016) Adherence to Concurrent Tuberculosis Treatment and Antiretroviral Treatment among Co-Infected Persons in South Africa, 2008–2010. *PLOS ONE* 11(7): e0159317. <https://doi.org/10.1371/journal.pone.0159317>

Wanyama JN, Nabaggala SM, Kiragga A, Owarwo NC, Seera M, Nakiyingi W, et al. High mobile phone ownership but low internet access and use among young adults attending an urban HIV clinic in Uganda. *Vulnerable Child Youth Stud.* 2018.

Ggita JM, Ojok C, Meyer AJ, Farr K, Shete PB, Ochom E, Turimumahoro P, Babirye D, Mark D, Dowdy D, Ackerman S, Armstrong-Hough M, Nalugwa T, Ayakaka I, Moore D, Haberer JE, Cattamanchi A, Katamba A, Davis JL. Patterns of usage and preferences of users for tuberculosis-related text messages and voice calls in Uganda. *Int J Tuberc Lung Dis.* 2018 May 1;22(5):530-536. doi: 10.5588/ijtld.17.0521. PMID: 29663958; PMCID: PMC6350252.

Frost H, Campbell P, Maxwell M, O'Carroll RE, Dombrowski SU, Williams B, Cheyne H, Coles E, Pollock A. Effectiveness of Motivational Interviewing on adult behaviour change in health and social care settings: A systematic review of reviews. *PLoS One.* 2018 Oct 18;13(10):e0204890. doi: 10.1371/journal.pone.0204890. PMID: 30335780; PMCID: PMC6193639.

An interesting observation is that some subjects did not receive their SMS messages. The explanation seems to be repetitive. What is the difference between being 'disconnected from the network', having a number that is deactivated, or a phone that is off for a long time? Phones that are off for a long time still receive the SMS eventually.

The explanation for SMS non-delivery was taken from the technical documentation of the SMS gateway used for this study: <https://guide.telerivet.com/hc/en-us/articles/360038959611-Message-Statuses>. The SMS gateway also provided the SMS data we used in our analysis. Our findings also seem to be consistent with the SMS delivery rates of MommConnect:

"Users received over 80% of expected MommConnect messages over time. Delivery rates were stable across provinces but differed over time and by MNO type. By MNO, Telkom (71%) and Cell C (75%) reported the lowest message delivery success rates versus Vodacom (81%) and MTN (82%).

Reasons for message delivery failure unfortunately were not available for all networks nor systematically assessed throughout the life of the program. However, based on the available data for

2016, the leading reason for non-delivery was that the SMS had expired—a likely indicator of an inactive phone number (online supplementary table 3).”

<https://www.ncbi.nlm.nih.gov/pmc/articles/PMC5922477/#SP1>

Also, although we could not confirm this, we suspect that some participants offered fictitious numbers or landline numbers in order to obtain reimbursement money.

Q: “Phones that are off for a long time still receive the SMS eventually.”

Off-record discussions with mHealth and SMS gateway companies suggest that mobile network providers (MNOs) attempt x (depends on the MNO) number of times to send a message, and then stop. The number of attempts depends on the MNO.

We nevertheless dropped our statement from the manuscript to reduce the word count in reworking the discussion to ensure a logical flow.

Discussion, para 2, lines 9-11: the switch to telephonic follow-up should be reported in the Methods.

This has been added, page 9

“During COVID-19 lockdown, we switched to telephonic follow-up of participants using a shortened questionnaire whereby only strictly needed information for the measurement of outcomes was inquired about.”

The study protocol was published, but with hindsight, was the study design appropriate? Having had a ‘no effect’ outcome, it would be helpful to identify possible weaknesses and shortcomings of the study design. The findings could be contextualised within the specific literature, both recent and past, on studies that did not show an effect of either motivational interventions and SMS messages in this patient population.

The finding that the intervention had no effect is unlikely to be due to limitations of the randomised control design, which in principle provides the strongest unbiased evidence of effectiveness. Alternative designs, such as a cluster randomised or non-randomised quasi-experimental trial, would be unlikely to have demonstrated a true effect which this trial might erroneously have failed to demonstrate. Cluster randomization was not necessary because there was in fact minimal or no contamination of control participants by the intervention, which could potentially have obscured a true effect in this individually randomised trial. A non-randomised design would be more likely to have produced a biased result.

However, ideally we would like to have a 4-arm design: control only, SMS only, MI only and SMS plus MI but available funds did not permit us to do so. We demonstrated that a single MI intervention was effective in doubling smoking cessation in TB patients in a very similar context in South Africa. There were theoretical reasons from the model published in our formative study to believe that the combination of smoking, drinking and adherence could impact TB treatment outcomes. Nevertheless, we deemed it necessary to increase the number of MI sessions and adding SMS because of the complexity of the intervention. We studied the feasibility of the intervention first and learnt valuable lessons about the MI training and follow-up (which we intensified in the RCT) and the potential pitfalls with SMS (which we also attempted to address).

As rightly pointed out, some new evidence was published recently. We contextualised our discussion in the light of the new findings, as follows:

Limitations: The 2-arm study design did not permit the untangling of the individual effects of SMS and MI. Knowing this could have important cost implications as SMS-communication would be cheaper and easier to organise than individual counselling.

Final paragraph of discussion: Lastly, in the light of the already existing evidence of SMS and the costs and implementation challenges relating to MI, intervention studies limited to an mHealth based intervention but using different intensities, duration and type of interventions (oneway, twoway, interactive) are needed.

Below is a larger section from the limitations, which was restructured extensively to capture concerns expressed by yourself and the other reviewer.

The 2-arm study design did not permit the untangling of the individual effects of SMS and MI. Understanding their separate effects could have important cost implications as SMS-communication would be cheaper and easier to organise than individual counselling. The lack of effectiveness of our intervention on the primary outcome (TB treatment success) can have a number of possible explanations. Although intervention uptake was high (80.2%) for the first counselling session, many participants did not return for the second (29.7%) and third (47%) sessions. As a result of this only about half of the intervention arm participants received all three MI sessions. Furthermore, about one quarter of all participants did not receive any SMS-messages. Low intervention uptake leads to a dilution of any potential effects. The lack of effectiveness on TB treatment success could perhaps also be explained by the complexity of the ProLife intervention itself: counsellors had to address multiple behaviours, namely medication adherence, tobacco smoking and hazardous/harmful drinking. Despite having established the feasibility and acceptability of this approach (29) and ongoing on site performance monitoring and feedback of counsellors by an MI expert, it is possible that MI for multiple behaviour change in the ProLife study was counterproductive as counsellors may have ended up not focusing on any of the behaviours at optimal levels. Similarly, patients might have found it difficult to change multiple behaviours simultaneously, especially because smoking and drinking are mutually reinforcing. This integrated approach was nevertheless adopted to avoid the need for multiple vertical counselling services (in addition to TB treatment and HIV treatment), to allow the different elements of the programme to reinforce one another, and to improve the affordability, feasibility and acceptability for a future roll-out of the programme. It is also possible that sequential interventions may be better, at least for smoking cessation.(42) More intensive counselling (more sessions) or a modified counselling method may have been more appropriate, even more so since a recent review of reviews of MI casts doubt on its efficacy.(43) For example, more emphasis on increasing patient knowledge in addition to increasing self-efficacy may have been more effective.(44) The cause of the mobile health message delivery problems (such as poor network coverage, no electricity to charge phones) (41) would need to be investigated in order to increase the effectiveness of future mHealth interventions. Messages may also have to be intensified or modified to be more interactive and/or tailored to specific circumstances of each individual. This would improve the personal value of the intervention to the individual, which is likely to increase the chances of their participation in the intervention.(45)

Reviewer: 2

Dr. Vijayan Shibu

Comments to the Author:

Appreciate the attempt to publish though the outcomes are not positive. This also reinforces that fact that habits and substance abuse need to be addressed as a disease or more rigorous interventions needed . Soft touch interventions may fall short.

Reviewer: 3

Dr. Himanshu Gupte, Narotam Sekhsaria Foundation

Comments to the Author:

Congratulations for a well written article especially when the results were not favourable!

Overall, it is a well written and well-explained article giving an insight into the complex intervention and challenges faced in its implementation.

While most aspects have been logically discussed in detail, I find some lack of clarity on the implementers' perspective. The LHWs who were the most important aspect of the intervention have been discussed superficially. This was a first trial with a complex intervention involving MI by LHWs. The discussion on probable reasons of the outcomes does not include their role. While there is discussion about whether the intervention should have been more intensive or some other technique should have been used, the assessment of the LHWs was based on only 20 minutes of interview. Is this sufficient to assess this? Further, there were some red flags in the feasibility study about failure to

reach MI threshold levels by LHWs and some implementation challenges were also mentioned by them. How was this addressed in the intervention?

Thank you for your useful feedback. The 20 minutes review is the standardised approach used in the Motivational Treatment Interview Integrity score. The instrument was validated and tested for reliability on this basis. Note that counts of certain counselling behaviours would be incorrect if we deviated from the standardised duration, i.e. using a longer duration would bias the number of reflections, affirmations etc.

W.r.t the red flags in the feasibility study, we addressed those in the trial as follows:

- The MI training was modified based on feedback received from the feasibility study.
- Privacy was increased with the purchase of Gazebo's for outside counselling
- A trained MI counsellor was appointed for the RCT to visit the sites on a regular basis, to discuss challenges relating to MI counselling and to do on-site training (roleplays) with LHCWs
- The number of MI counsellors was increased to avoid that counsellors had to travel long distances to different clinics, to avoid that counsellors were not present when a patient with TB came for counselling. This change was implemented after the trial has started, on receiving feedback from counsellors regarding this.

The following changes were made to the manuscript (discussion, paragraph 2 and limitations)

The results of our MI analyses suggest that the LHWs trained as counsellors were more proficient in MI than during the feasibility stage, as observed by their global rating scores on cultivating change talk, softening sustain talk, partnership and empathy. (Supplementary Table 4). These results were achieved by ongoing monitoring and training of LHWs during the trial and adapting the training to feedback received from the feasibility stage. Extra counsellors were also appointed to minimise travel distances to clinics.

Despite having established the feasibility and acceptability of this approach (29) and ongoing on site performance monitoring and feedback of counsellors, it is possible that MI for multiple behaviour change in the ProLife study was counterproductive as counsellors may have ended up not focusing on any of the behaviours at optimal levels.

Another question is that when the intended intervention could not fully implemented due to various reasons discussed, is it appropriate to say that it was not effective?

Thank you for your valuable feedback.

The earlier termination date of the recruitment should not have impacted on our anticipated power to detect a difference for the primary outcome for which the LTFU was only 13.4 %, much lower than the anticipated 25%. If we calculate the power to detect a difference of 10% (proportions of 0.86 in the intervention vs 0.76 in the control group) and use our actual achieved sample size of 283 in the intervention group, then the calculated power to detect a difference is 0.83. This is in fact slightly higher than our desired power of 0.80. Our conclusions regarding the primary outcome are therefore not influenced by the lower than planned sample size.

However, it is certainly correct that the smaller final sample size impacted the power to detect a difference for secondary outcomes where the LTFU was much higher than the anticipated 25%.

It is also important to note that the sample size was not powered for these secondary outcomes which – with the exception of adherence information-only applied to subgroups of drinkers or smokers or HIV-positive people or patients diagnosed based on positive bacteriology. The latter has been added as a limitation of the study.

The low uptake of the intervention may also have diluted effects.

We therefore reworked the discussion, particular the limitations and the conclusions and the flow of the text in several places. The most important changes are as follows:

-First paragraph of introduction: We could also not demonstrate significant beneficial effects on any of the secondary outcomes, i.e., smoking, alcohol consumption, medication adherence and ART initiation.

-Limitations. Trial recruitment had to be terminated before the planned sample size because of funding and time constraints. Nevertheless the calculation of sample size was based on an anticipated 25% LTFU for the primary outcome, while in reality only 13.4 %, of the TB outcomes were not available. As a result, we achieved a slightly higher power to detect the a 10% difference in primary outcomes, than we had aimed for (83% vs. 80%). The smaller sample size did however reduce the power to detect a difference for secondary outcomes for which the LTFU was much higher than 25%. Furthermore the calculated sample size was not powered for subgroups analysis which was the case for outcomes relating to smoking, drinking, ART and cure rates. Furthermore, loss to follow-up (LTFU) was high for the 3- and 6-month questionnaires despite all possible efforts to increase follow-up, thus seriously impacting power to detect a difference in secondary outcomes. In addition, due to COVID-19 lockdown in March 2020, we had to switch to telephonic follow-up of participants using a shortened questionnaire (22 participants) and could not access clinics to retrieve outstanding TB treatment outcomes. The low intervention uptake meant that half of the participants received only one or two MI sessions combined with SMS messages. SMS messages were only used for the first half of the study period, and one quarter of participants did not receive their messages. It could be argued that in the absence of ongoing text messages, the motivational interventions and associated text messages were not enough to keep participants focussed for the second three months of the trial.

In conclusion, we could not demonstrate that the ProLife intervention was effective in improving TB treatment outcomes. This may be due to the lack of effect of the intervention on the intermediary secondary outcomes, but the study may also have been underpowered to detect significant differences, particularly in secondary outcomes. Valuable lessons were learnt on challenges relating to training LHWs in MI counselling and delivery, SMS-delivery in a challenging socio-economic context and the reasons for loss to follow-up of TB participants with multiple health problems. Further research is needed to provide answers on how to increase intervention uptake in poor resource settings and whether our complex intervention should have been more intensive. Other important questions are whether another counselling method would have been more effective. Lastly, in the light of the already existing evidence of SMS and the costs and implementation challenges relating to MI, intervention studies limited to an mHealth intervention but using different intensities, duration and type of interventions (one-way, two-way, interactive) are needed.

The article is worth publishing for dissemination of these interesting findings and learnings but some more light has to be shed on these aspects pointed out here.

VERSION 2 – REVIEW

REVIEWER	Mars, Maurice University of Kwazulu-Natal, TeleHealth
REVIEW RETURNED	10-Dec-2021
GENERAL COMMENTS	Thank you for the opportunity to review the revised version of the paper. The authors' responses have helped clarify my concerns about the study and it design. At best, only 53% of people in the

	study group received the full intervention, three motivational interventions and 24 SMS messages, and 25% did not receive any messages. The previous comment about a possibly flawed study design remains. The issue was not about a randomised controlled study but the factors that were taken into account in determining the sample size. The sample size was calculated on an assumption of 25% of people lost to follow up. The authors maintain that even though they did not achieve the predicted sample size, this was not a concern as only 13% of the sample were lost to follow up. What was not taken into account in the design was the percentage of people who would not receive their SMS messages and the percentage who would not return for their motivational intervention. There are data from other studies that could have guided the sample size prediction. It is not clear whether people only received SMS messages if they attended the motivational interventions or of all those enrolled were meant to receive all 24 messages. Of the 53% of people who received three motivational sessions, what percentage received all the messages or none of the messages? It would be helpful if the results could be disaggregated to some extent as trends might emerge. What were the outcomes for those who received the full intervention and those who received only one motivational intervention and some SMS, or those who had one motivational intervention and no SMS messages? The first motivational intervention involved prioritisation and agenda setting. Those who received only one session would not have had all their issues addressed. The implication is that these people would have had no intervention other than perhaps SMS messages for one or two of the factors. Again, what useful information can be gained from disaggregating the data to see if a single motivational intervention had any effect on the topic addressed? This might, for example, help in understanding why the authors were unable to demonstrate the reduction in smoking that they found in a previous study. How many people received a motivational intervention related to smoking?2:33 PM 1/7/2022
--	--

VERSION 2 – AUTHOR RESPONSE

Thank you for the opportunity to review the revised version of the paper. The authors' responses have helped clarify my concerns about the study and its design. At best, only 53% of people in the study group received the full intervention, three motivational interventions and 24 SMS messages, and 25% did not receive any messages.

The previous comment about a possibly flawed study design remains. The issue was not about a randomised controlled study but the factors that were taken into account in determining the sample size. The sample size was calculated on an assumption of 25% of people lost to follow up. The authors maintain that even though they did not achieve the predicted sample size, this was not a concern as only 13% of the sample were lost to follow up. What was not taken into account in the design was the percentage of people who would not receive their SMS messages and the percentage who would not return for their motivational intervention. There are data from other studies that could have guided the sample size prediction.

A priori sample size estimation for a given effect size takes account of potential attrition rates but not treatment compliance rates. Of course, a larger sample size might have been able to detect a smaller effect size but we designed our trial to detect an effect size of a specific magnitude, which we considered clinically significant. The fact that this specific effect size was not detected was not because we had an insufficient sample size but because the intervention did not produce an effect of the desired magnitude; we accept that the poor compliance was partly responsible for this lack of effectiveness of the intervention in this pragmatic trial. Note that lack of uptake of an intervention is also a finding in a pragmatic effectiveness trial: it is even more unlikely that the intervention would be effective in real life situations than in the ideal settings of a trial.

It is not clear whether people only received SMS messages if they attended the motivational interventions or of all those enrolled were meant to receive all 24 messages.

SMS-messages were automatically activated after the first MI had taken place. Thereafter all messages were delivered even if the participant did not attend the second or third MI session. It would have been very difficult to set up the system differently as we were not certain that a person would attend a second session until he or she had or had not turned up. Note though that participants who only smoked but did not drink harmful or hazardous extents received 17 messages and not 24 (i.e. 10 adherence messages and 7 smoking related messages) and the other way round. This is explained in more detail in the feasibility paper. Word length of the current paper did not permit detailed explanation in the RCT paper. However, we added the following statement to the paper:

“SMS-messages were automatically activated after the first MI had taken place. Thereafter all applicable messages were delivered even if the participant did not attend the second or third MI session.”

Of the 53% of people who received three motivational sessions, what percentage received all the messages or none of the messages? As you can see from the table below 43% (64 out of 150 participants who received all 3 MI sessions) received all applicable messages. (applicable refers to the fact that they would by definition not receive alcohol related messages if they were not hazardous or harmful drinkers, or not receive smoking related messages if they were not current smokers at enrolment). This is not very different from the percentage of participants who received all applicable messages after receipt of MI1 (95/227 (41.9%))

% SMS-messages received for the group who received all 3 MI counselling sessions	n(%), N = 150
p_del_cat	
0%	14 (9.3%)
1% to 20%	5 (3.3%)
21% to 40%	5 (3.3%)
41% to 60%	10 (6.7%)
61% to 80%	11 (7.3%)

% SMS-messages received for the group who received all 3 MI counselling sessions	n(%) N = 150
81% to 99%	40 (27%)
100%	64 (43%)
More than 100%	1 (0.7%)

It would be helpful if the results could be disaggregated to some extent as trends might emerge. What were the outcomes for those who received the full intervention and those who received only one motivational intervention and some SMS, or those who had one motivational intervention and no SMS messages? The first motivational intervention involved prioritisation and agenda setting. Those who received only one session would not have had all their issues addressed. The implication is that these people would have had no intervention other than perhaps SMS messages for one or two of the factors. Again, what useful information can be gained from disaggregating the data to see if a single motivational intervention had any effect on the topic addressed? This might, for example, help in understanding why the authors were unable to demonstrate the reduction in smoking that they found in a previous study. How many people received a motivational intervention related to smoking?

We acknowledge that the proposed analyses would be interesting but these are post-hoc analyses and were not part of the protocol and hence are not conventionally reported in the main trial paper. The CONSORT guidelines state "Because of the high risk for spurious findings, subgroup analyses are often discouraged.....Post hoc subgroup comparisons (analyses done after looking at the data) are especially likely not to be confirmed by further studies. Such analyses do not have great credibility." (<http://www.consort-statement.org/checklists/view/32--consort-2010/97-additional-analyses>). One of the important reasons for this is that the roughly equal distribution of potentially confounding factors achieved by randomisation is violated by these additional subgroup analyses, thus explaining those spurious associations.

(Also note that even if we did subgroup analyses against the recommendations of the Consort guidelines, they would have to be done with even smaller numbers than in the actual trial, thus making statistical comparisons inappropriate with very wide confidence intervals. For example only 22 smokers quit in the entire trial: numbers would be even lower by limiting the analysis to those who received the corresponding MI session. This is however not the main reason for not performing those analyses, as small numbers would affect precision while the real problem is potential bias introduced by subgroup analysis)

VERSION 3 – REVIEW

REVIEWER	Mars, Maurice University of Kwazulu-Natal, TeleHealth
REVIEW RETURNED	29-Dec-2021
GENERAL COMMENTS	My comment related to disaggregating the data was not necessarily about showing statistical difference but rather about noting trends that might warrant further investigation.